# Multimethods study comparing the experiences of medical clinical academics with nurses, midwives and allied health professionals pursuing a clinical academic career

Diane Trusson [ID],[1] Emma Rowley,[1] Jonathan Barratt[2]

► Prepublication history and additional supplemental materials for this paper is available online. To view these files, please visit the journal online (http://dx.doi.org/10.1136/bmjopen-2020-043270).

[1]School of Medicine, NIHR Applied Research Collaboration East Midlands (ARC EM), University of Nottingham, Nottingham, UK
[2]College of Life Sciences, University of Leicester, Leicester, UK

**Correspondence to**
Dr Diane Trusson;
diane.trusson@nottingham.ac.uk

## ABSTRACT

**Objectives** This study aimed to compare experiences of medical clinical academics (MCAs) with those of nurses, midwives and allied health professionals (NMAHPs) pursuing a clinical academic career.

**Design** A multimethods approach was used to elicit qualitative data. Both sets of participants completed similar online surveys followed by in-depth interviews to explore emerging themes.

**Setting** The research was conducted in the East Midlands of England, encompassing two Higher Education Institutions and four National Health Service Trusts.

**Participants** Surveys were completed by 67 NMAHPs and 73 MCA trainees. Sixteen participants from each group were interviewed following a similar interview schedule.

**Results** The survey data revealed notable differences in demographics of the two study populations, reflecting their different career structures. MCAs were younger and they all combined clinical and academic training, lengthening the time before qualification. In contrast, most NMAHPs had been in their clinical post for some years before embarking on a clinical academic pathway. Both routes had financial and personal repercussions and participants faced similar obstacles. However, there was also evidence of wide-ranging benefits from combining clinical and academic roles.

**Conclusions** Variations in experiences between the two study populations highlight a need for a clear academic pathway for all health professionals, as well as sufficient opportunities post-PhD to enable clinical academics to fully use their dual skills.

## INTRODUCTION

Recognising the value of supporting clinicians to do research in their area of expertise, the National Institute for Health Research (NIHR) provides a range of schemes to encourage aspiring clinical academics.[1] For medical clinical academics (MCAs), the academic career path can begin at medical school with an intercalated degree or bachelor of medicine PhD programme, alongside other opportunities to be involved in

### Strengths and limitations of this study

► This is the first study to compare experiences of medical clinical academics with those of nurses, midwives and allied health professionals pursuing a clinical academic career.
► A multimethods approach enabled both breadth and depth of data to be gathered for comparison purposes.
► The study was limited to one geographical area.

research. Academic foundation programme posts funded by Health Education England (HEE), offer protected academic time during foundation year 2. The NIHR Integrated Academic Training (IAT) Pathway supports academic clinical fellowships (ACFs) enabling academic training alongside specialty training with the aim of supporting entry onto a PhD programme, doctoral research fellowships and clinical lectureships (CLs) enabling postdoctoral clinicians to split their time equally between clinical and academic work [2] (see figure 1).

In parallel, the HEE and NIHR Integrated Clinical Academic Programme provides research training awards for healthcare professionals (excluding doctors and dentists) wanting to develop careers combining clinical research and research leadership with continued clinical practice and professional development (see table 1).[3] In contrast to MCAs, the nurses, midwives and allied health professionals (NMAHP) clinical academic pathway awards can only be pursued postgraduation and on completion of clinical preceptorship.

For both professional groups, benefits of pursuing a clinical academic career include intellectual stimulation and improved career prospects[4 5] as well as wide-ranging benefits for

**BMJ**

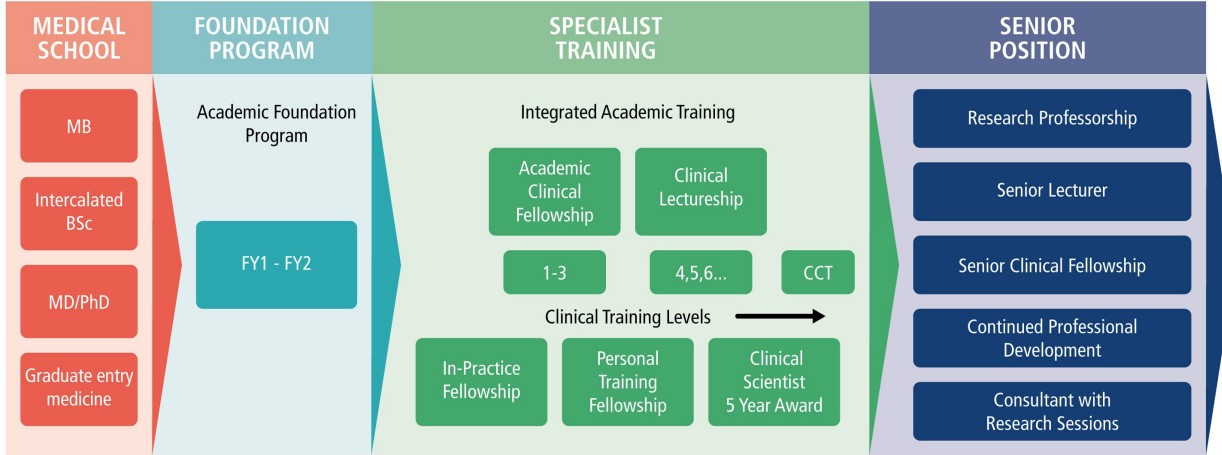

**Figure 1** Medical clinical academic progression. CCT, certificate of completion of training.

patient care.[6] Studies also identify challenges, including unsupportive work environments, competing pressures, scarcity of funding and senior academic appointments and difficulty achieving a good work/life balance.[7 8] Previous studies have focused either on MCAs,[9–11] or on NMAHPs.[12] However, it remains unclear how the benefits and challenges associated with the clinical academic pathway compare between MCAs and NMAHPs.

In addition, concerns about a 'decline in the capacity of National Health Service (NHS) staff to undertake, or even engage with research'[6p.6] as well as rates of attrition along the clinical academic career trajectory, particularly of women,[6 13] have made it increasingly important to identify factors that encourage and enable continuation to higher levels of clinical academia.[7]

This paper reports on data from two studies conducted in the East Midlands: one with NMAHPs, and one with MCAs. The aim is to compare experiences of pursuing a clinical academic career between the two study populations.

## METHODS
This manuscript has been prepared according to the Standards for Reporting Qualitative Research[14] (see online supplemental file 1).

### Study design
A qualitative methodology was used for this study in order to prioritise individual perspectives and allow themes to arise from the data.[15] The study had two data collection components. First, an online survey gathered descriptive data from a relatively large number of respondents at different stages along the clinical academic pathway. This was followed by semistructured interviews to enable a deeper exploration of individual experiences which can be lost in focus group interviews for example.[10]

### Recruitment
Participants were recruited from two main medical teaching Higher Education Institutions (HEIs) and four large NHS Trusts in the East Midlands.

Half of each survey's respondents indicated their willingness to be interviewed. Quota sampling was used[15] to select 16 respondents from each study population, representing a range of demographics and stages of training. See tables 2 and 3 for interview sample characteristics.

### Data collection
An academic online survey platform (JISC Online Survey)[16] was used in each study (see online supplemental files 2 and 3).

| Table 1 | HEE/NIHR Integrated Clinical Academic Programme | | | |
|---|---|---|---|---|
| **Internships** | **Masters' in clinical research studentships** | **Clinical doctoral research fellowships** | **Clinical lectureships** | **Senior clinical lectureships** |
| 'A premasters clinical research taster' | 'An introduction to clinical research theory and practice' | 'Obtain a PhD by research while still developing clinical skills' | 'Combine postdoctoral research in an academic position with continued clinical practice' | Combine research and research leadership in a senior academic position with continued clinical practice' |

NIHR, National Institute for Health Research.

**Table 2** NMAHP interview sample characteristics

| NMAHP case study number | Clinical role | Stage of study (at interview) | Age group | Gender |
|---|---|---|---|---|
| NMA1 | Nurse | Thesis pending | 20–30 | Female |
| NMA2 | Nurse | Year 2 PhD | 41–50 | Female |
| NMA3 | Nurse | Year 4 PhD | 41–50 | Female |
| NMA4 | Midwife | Post-doc | 41–50 | Female |
| NMA5 | Nurse | Year 3 PhD | 41–50 | Female |
| NMA6 | AHP | Year 4 PhD | 41–50 | Female |
| NMA7 | AHP | Year 2 PhD | 41–50 | Female |
| NMA8 | Midwife | Year 5 PhD | 41–50 | Female |
| NMA9 | Nurse | Post-doc | 31–40 | Male |
| NMA10 | AHP | Year 2 PhD | 41–50 | Female |
| NMA11 | AHP | Year 1 PhD | 41–50 | Female |
| NMA12 | AHP | Year 4 PhD | 51+ | Male |
| NMA13 | AHP | Year 3 PhD | 31–40 | Male |
| NMA14 | AHP | Year 3 PhD | 41–50 | Male |
| NMA15 | AHP | Post-doc | 51+ | Male |
| NMA16 | AHP | Year 1 PhD | 31–40 | Male |

AHP, allied health professionals; NMAHP, nurses, midwives and allied health professional.

The survey questions were developed with reference to previous literature[5 7 10] and aimed to build on existing knowledge regarding trainees' experiences of clinical

**Table 3** MCA interview sample characteristics

| MCA case study no | Clinical training programme (at interview) | Age group | Gender |
|---|---|---|---|
| Med1 | ACF | 31–40 | Female |
| Med2 | Clinical lecturer | 31–40 | Female |
| Med3 | Year 3 PhD | 20–30 | Female |
| Med4 | ACF | 20–30 | Male |
| Med5 | AFP | 20–30 | Male |
| Med6 | AFP | 20–30 | Male |
| Med7 | Year 2 PhD | 31–40 | Male |
| Med8 | ACF | 31–40 | Female |
| Med9 | Year 4 PhD | 31–40 | Male |
| Med10 | Clinical lecturer | 31–40 | Female |
| Med11 | ACF | 20–30 | Male |
| Med12 | Clinical lecturer | 31–40 | Female |
| Med13 | Clinical lecturer | 31–40 | Female |
| Med14 | Clinical lecturer | 31–40 | Male |
| Med15 | Year 2 PhD | 31–40 | Female |
| Med16 | ACF | 31–40 | Female |

ACF, academic clinical fellowships; AFP, Academic foundation programme; MCA, medical clinical academic.

academic training and their career aspirations. The interview guides were developed to allow deeper exploration of individual experiences, following themes which arose in the surveys.

Both sets of participants were given the same set of survey and interview questions with the exception of questions relating to their specific circumstances, such as asking for participants' medical specialty rather than job title.

Furthermore, the MCA survey, which was created after the NMAHPs' data had been analysed, contained an additional question about anticipating challenges, such as taking parental leave or working part-time in the future. In interviews, MCAs were asked for their perspectives on combining clinical academic careers with family life. These themes arose unsolicited in the NMAHPs' interviews and were considered important to pursue in a comparative study because of concerns around diversity, particularly at higher levels[13] and suggestions that difficulty in achieving a good work-life balance could be a barrier to clinical academic career progression.[7]

A link to the survey was distributed through a clinical academic network for NMAHPs[17] and a database of 263 MCA trainees. Both surveys were open for approximately 2 months. Potential participants had the opportunity to enter a prize draw as an incentive to participate.

Acknowledging clinical academics' limited availability, individual interviews were conducted at a time and place to suit the interviewee. This was usually their place of work or a public place (eg, café). All interviews began by asking about their experiences of being a clinical academic, enabling the participant to prioritise what they considered to be the most salient aspects of their experiences. Although the interviews were flexible, a preprepared list of questions enabled responses to be compared within the sample and also between research populations.[18]

All interviews were digitally recorded with the participants' permission.

## Research team
The team consisted of a medical sociologist, a capacity development manager and a clinical academic from the respective study population (a nurse in the NMAHP study and a surgeon in the MCA study) to help tailor the survey and interview questions and facilitate participant recruitment. The first author who conducted all the interviews, was an experienced social researcher with no prior relationship with the participants.

## Ethical considerations
Following Health Research Authority[19] guidance, this research was not submitted for Ethics Board approval because participants were recruited by virtue of their participation in educational programmes, not their NHS status. Nevertheless, the research was conducted in accordance with the Declaration of Helsinki.[20] Any identifying information given in order to enter the prize draw and/or offer to be interviewed, was removed prior to analysis.

 3

Interviewees were informed about the purpose of the research, their right to withdraw from the study and how their (anonymised) data would be used. Recorded verbal consent was obtained prior to each interview. When interviewing in a public space, privacy was ensured to maintain confidentiality.

In addition, participants were assured that data would be stored confidentially on secure University systems.

### Data processing
Demographic descriptive data from the surveys were summarised in graph form and free-text responses were collated using JISC software.[16] Interviews were transcribed professionally with all identifying data removed prior to analysis. To ensure anonymity, NMAHP survey respondents are identified as NMASR1-SR67 or case study number (NMA1-16) plus their role. MCA survey respondents are identified by MedSR1-73, or case study number (Med1-16) with their stage of training. In addition, data extracts used in the discussion were sent to the relevant participants to confirm that they could not be identified by the contents.

### Analysis
A framework analysis was used as it is an effective way of reducing and summarising large amounts of qualitative data. It is appropriate for this type of study where 'it is important to compare and contrast (textual) data by themes across many cases'.[19 p.6] The first author conducted the initial coding through an inductive approach involving multiple readings of the survey responses and interview transcripts. The resulting codes were compiled into a set of potential themes[20] which were discussed and agreed with the coauthors to ensure reliability.[15] Furthermore, combining survey and interview data increased the credibility of the findings.[15]

D

### Patient and public involvement
No traditional patient/public involvement was included in this study; however, the clinical members of the author team were able to act as sounding boards for their respective communities, to ensure data collection tools and plans were appropriate for the clinical communities being explored.

### RESULTS
The surveys were completed by 67 NMAHPs and 73 MCAs who described their job role (figure 2) or stage of training (figure 3) as follows.

### Gender
Figure 4 reveals relatively equal numbers of male (41) and female (32) MCAs survey respondents, whereas female NMAHPs outnumbered their male counterparts by 5:1. These figures are representative of the gender divide for these occupations nationally.[21] However, it is worth noting that these data were gathered from MCAs at an early stage

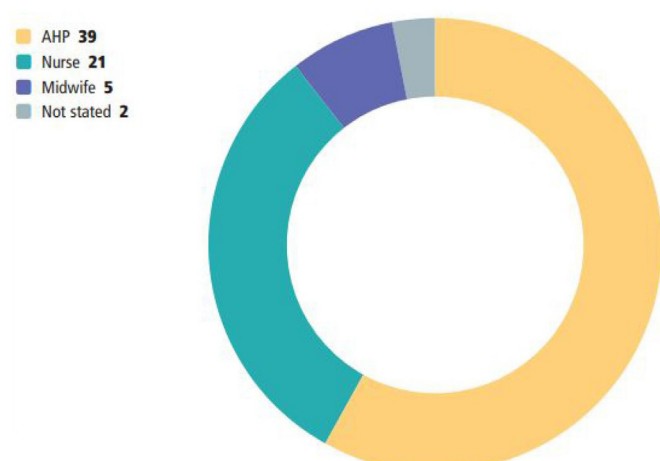

**Figure 2** NMAHP's profession. AHP, allied health professional; NMAHPs, nurses, midwives and allied health professionals.

in their careers and 'although approximately 50% of predoctoral ACFs are female…the percentage of female fellows declines with increasing seniority of award'.[1p 1]

### Age
There were major differences between the two populations in terms of age of the survey respondents:

Figure 5 reveals substantially higher numbers of MCAs in the 20–30 age group compared with NMAHPs. This reflects the career structures of the two populations. With medical trainees being encouraged to undertake academic training in parallel with their clinical training, they would be more likely to fall into the 20–30 bracket.

Although the number of NMAHPs aged 51+ was not high, it contrasted with the lack of any MCAs in this age group. This is consistent with findings which revealed that NMAHPs tend to commence the clinical academic pathway after establishing their careers whereas MCAs practising regionally who are 50+ are no longer on the

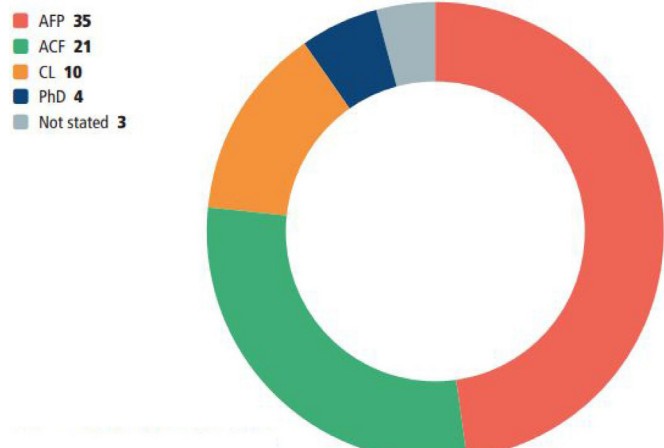

**Figure 3** MCA stage of training. ACF, academic clinical fellowships; AFP, academic foundation programme; CL, clinical lectureship; MCA, medical clinical academic.

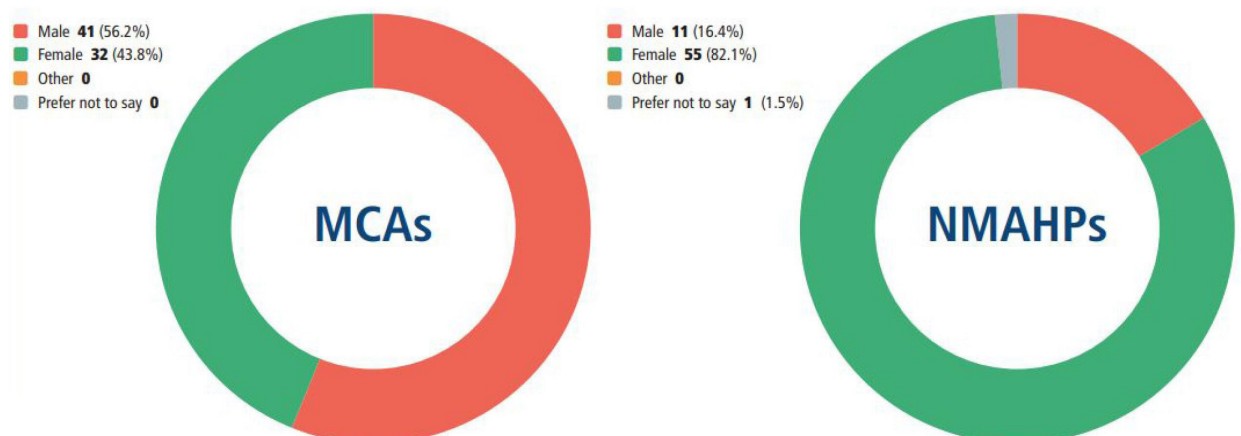

**Figure 4** How participants described their gender. MCAs, medical clinical academics; NMAHPs, nurses, midwives and allied health professionals.

training pathway; hence they are not captured in this dataset.

In the following section, the themes which arose in the data (see figure 6) will be discussed, with the exception of the theme 'combining clinical academia and family life' where the richness of the data was such that it will be reported in full separately.

### Embarking on a clinical academic career

In contrast to medical trainees who are encouraged to embark on the clinical academic pathway early in their career, academic training was a step-change in NMAHPs' career trajectory. This seemed to lead to a difference in attitudes towards participants in the two studies. For example, younger NMAHPs had to overcome certain prejudices:

> We've got this traditional sort of snobbery that you have to serve X amount of years before you are worthy to do academia (NMA8 Midwife, female).

This reveals how NMAHPs might be discouraged from pursuing a clinical academic career at an early stage, potentially holding talent back.[11]

### Motivation

Both sets of participants described being motivated to do research by a drive to improve patient care:

> What got my brain ticking was why one patient presented a certain way, and not another patient with exactly the same disease. I always wanted to know the why (Med2 Clinical lecturer, female).

However, findings revealed the range of research made possible by approaching it from different perspectives. For example:

> We don't want to just hand [research] over to medics. Because they see the women who are ill or need help, but we also need to improve care for the women who go through a normal pathway (NMA4 Midwife, female).

This is supported by the Dean of the NIHR, Professor Dave Jones:

> We need to back away from the doctor-driven research narrative…[nurses, midwives and AHPs] are the experts in what they do—they must be supported

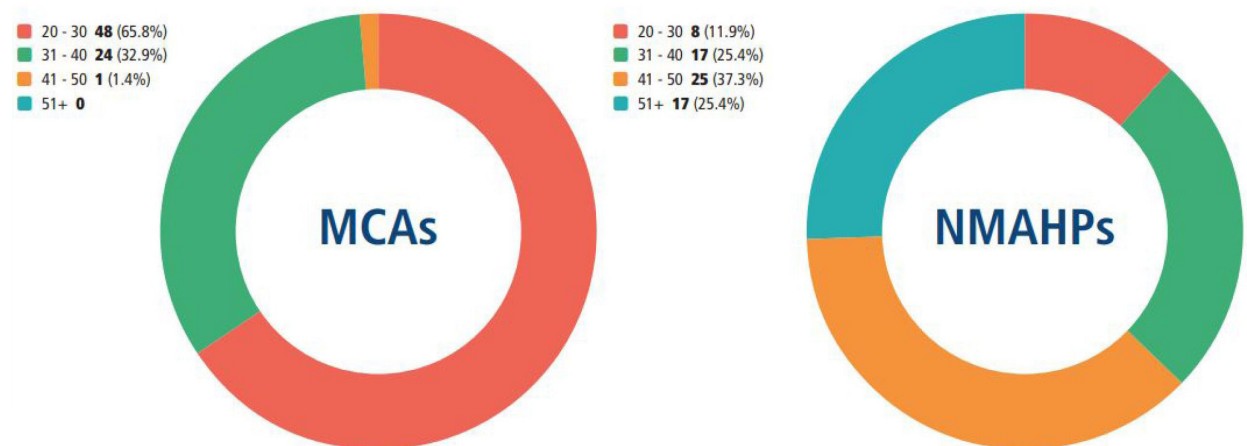

**Figure 5** Age groups of survey respondents. MCAs, medical clinical academics; NMAHPs, nurses, midwives and allied health professionals.

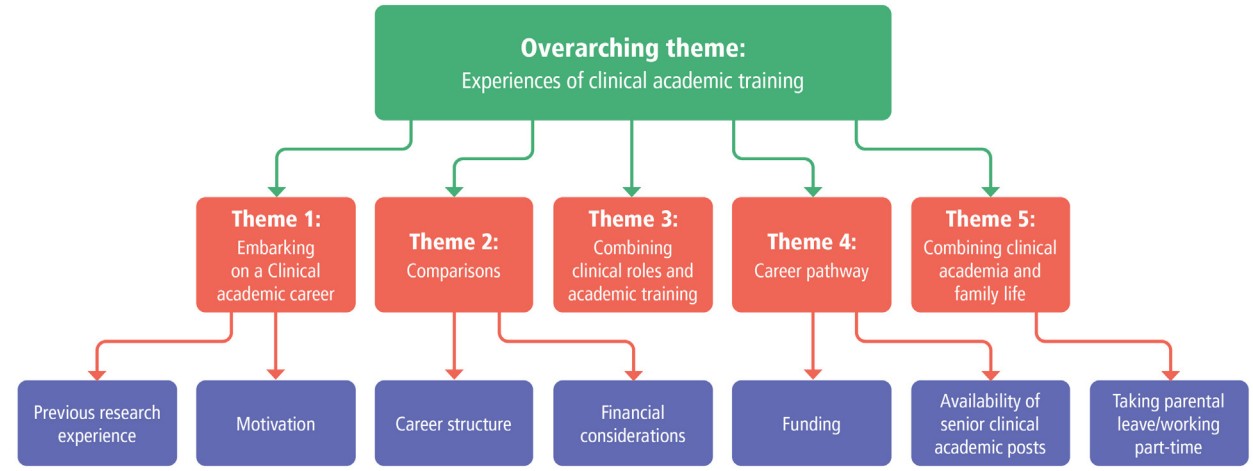

**Figure 6** Coding tree.

to do research because they are the ones who are seeing the gaps, and who know the questions that must be answered.[22]p.3

Another motivating factor for many MCAs was their early experiences of research:

> When I was a medical student [I took] one year out to do full-time research and I loved it (Med9 PhD, male).

This indicates the benefit of encouraging research, starting at undergraduate level.[6] In contrast, NMAHPs' prior exposure to research training varied according to their profession. For example, research has been part of the curriculum for student physiotherapists for many years, whereas nurses have only comparatively recently been required to undertake degree-level training.[23] This may explain why AHPs outnumbered nurses in the NMAHP study although changes may occur as more nurses emerge from training having had research experience.

However, while NMAHPs described choosing their research topic, the data revealed inconsistencies in the opportunities for MCAs to pursue their own personal research interests. One participant described the process for assigning research topics:

> We all put our names in a hat…I ended up with my third or fourth choice (Med3 ACF, Female).

Although this practice was only reported by participants from one training provider, it had the potential to deter people from doing research. For example, one ACF described being assigned a project which was not achievable in the 9 months available. Consequently, she had decided to focus solely on clinical training while her family were young.

> I haven't regretted doing it but…I haven't enjoyed it enough to keep going with it while I'm bringing up a family (Med1 ACF, female).

Similarly, several female NMAHPs described waiting until their children reached school age before embarking on a clinical academic pathway:

> My youngest daughter started school, so I had a bit of head space to think about where I wanted to take my career (NMA2 Nurse, female).

### Comparisons with MCAs

NMAHPs expressed frustration at what they perceived to be an 'easier' route for MCAs:

> For them it's a perfectly normal part of your career to do a PhD…it's such a shame that the culture is slow to be absorbed into the nursing profession (NMA5 Nurse, female).

> A doctor I've been working with was doing an MD… He said it was more prestigious to do a PhD, so they've just swapped him. Do you know the pains that I went through to get PhD funding and you've just swapped? (NMA11 AHP, female).

This participant seemed particularly unhappy by what was perceived as unfair financial penalisation:

> [Doctors] are given 2–3 days a week to do research, plus travel and parking expenses. My department doesn't have a training budget anymore. Any extra training you get, you pay out of your own pocket; you don't get study leave, you have to take annual leave. It's not equitable is it?' (NMA11 AHP, female).

The benefits of this support were confirmed by MCAs:

> I've been to lots of conferences…I've met a lot of influential people (its) really beneficial, for creating opportunities for my future (Med3 PhD, female)

Although the MCA route was better designed and funded, nonetheless MCAs faced stiff competition for posts (eg, the NIHR funds around 250 ACFs and 100 CL posts each year).[24] The following responses were typical:

The biggest challenge is getting funding/grants/fellowships that are limited (MedSR62 ACF, female).

There's no shortage of applicants; the problem is not having enough funding to go around. It's very difficult to get one of these posts. (Med11 ACF, male).

A particular concern of MCAs was how long it took them to qualify compared with colleagues who had chosen a purely clinical route:

All my peers who started with me are looking at Consultant jobs now and I've still got almost three years left (Med14 Clinical lecturer, male).

Nevertheless, participants also valued the length of training, seeing it as an opportunity to accumulate skills and knowledge:

The breadth of my clinical experience is a lot more than my clinical colleagues…I've seen more complications, more rare stuff, just because I've been around longer (Med10 Clinical lecturer, female).

### Financial considerations

The delays in clinical qualification challenged some NMAHPs' perception that pursuing a clinical academic career had 'no impact on medics' money':

It has delayed my CCT [Certificate of Completion of Training] and substantially reduced my life-time earnings (MedSR5 Clinical lecturer, female).

This was exacerbated for MCAs who took parental leave and/or worked part-time:

It really extends my training…that's five years of earning a Registrar level salary, rather than a Consultant one, and that's a big difference (Med15 Clinical lecturer, female).

In contrast, some NMAHPs who had progressed to high levels professionally before embarking on the clinical academic pathway, reported a drop in their earnings:

When the medics do their on-call shifts they're not asked to downgrade in any way but…I was expected to work as a Band 5, and I've been Band 7 for twelve years. You're trying to push yourself forward academically but clinically it's not respected (NMA2 Nurse, female).

Again, contrasts were drawn with MCAs:

You get the impression that it was much more set up for [MCAs] to be able to continue in research…they had an established career structure. Whereas in physiotherapy it's not the norm, it's massively the exception that involves you seeking a way of doing it. There isn't that pathway as part of the profession (NMA14 AHP, male).

### Combining clinical practice and academic training

Both sets of participants expressed difficulties in convincing managers of the value of academic training. For example:

The idea that you want to contribute in a different way is a bit like, 'Why? We need midwives at the frontline, not doing PhDs.' And that's such a short-sighted approach to the workforce. Because it's the opposite you see in the medics (NMA8 Midwife, female).

Despite this perception, MCAs also reported challenges through doing academic training in parallel with clinical training:

We had to have around 25 hours [for] all the different study days. I really had to fight for them, I had to take annual leave, I had to reiterate again and again… Although there was lots of support from the academic side, there didn't really seem to be an understanding of what an academic trainee was (Med13 Clinical lecturer, female).

This suggests a lack of awareness around clinical academic careers within clinical settings, although according to the interviewees this varied between specialties. Nevertheless, it highlighted the need for improved communication between HEIs and NHS partners:

You're working with [both] the hospital and the university so [it would be good for] the rota coordinators to be aware that you have academic commitments (Med4 ACF male).

Despite the recommendation that 'both clinical and research training are properly ring-fenced',[3, p21] participants from both studies reported difficulties negotiating time for academic work. However, experiences varied depending on research funders. Participants funded by the NHS reported feeling reluctant to assert their right to time off for academic training, whereas those funded by NIHR seemed better able to assert themselves:

That's the nature of the NIHR fellowship, it's 100% [whereas] when the department are paying some of the salary, they do expect some work for that (NMA13 AHP, male).

### Career pathway

While MCAs are either employed by a hospital trust with an honorary university contract or vice-versa,[1] there is currently no nationally agreed approach to contracts for NMAHPs. Consequently, some NMAHPs are employed by both organisations on separate contracts which often adversely affects their terms and conditions:

Because it's not ingrained within our career structures, you're sort of chopped up into oh, well half of you will work in clinical practice and half will work in academia (NMA9 Nurse, male).

In addition, because PhDs are not part of the NMAHPs' career pathway, many were unable to retain their clinical post:

> I've had to step out of my area of expertise…I had to resign from my job that I'd been doing eighteen years to take on the PhD, which I wasn't impressed by…I asked to have a secondment, but they [NHS] won't let you (NMA10 AHP, female).

Some NMAHPs were anticipating problems in pursuing a clinical academic career post-PhD.

> I can't see [employer] saying we value your research enough to change your job description…to a clinical academic. A lot of people in my position just go back to their clinical role (NMA14 AHP, male).

Despite NMA14's research having had a major impact with changes implemented nationally, he still faced an uncertain future in relation to his specialist job role. Participants described issues at a structural level:

> It's as if the higher-level managers are saying 'yeah we need this', but they're not facilitating the managers to facilitate the clinicians. If we look at the medics, they've got these clinical academic posts firmly in place. And we need to have a firmly established pathway (NMA15 AHP, male).

However, both sets of participants expressed uncertainty about their post-doctoral prospects. Concerns around competition for limited funding and jobs were cited multiple times in both surveys and interviews:

> My frustration [is that] the pathway is a pyramid therefore some people will not progress up (NMA12 AHP, male).

This 'pyramid' analogy was considered particularly problematic for female clinical academics:

> The jobs are fewer and fewer the higher you get. A lot of women start off [clinical] academic and then leave along the way because there aren't enough jobs. Whereas if you're single and you've got no responsibilities you can move all over the country (Med11 ACF, male).

## DISCUSSION

This is the first study to combine experiences of MCAs with NMAHPs pursuing a clinical academic pathway. Data revealed differences in demographics between the research populations. Most MCAs were younger, highlighting that for medical trainees combining research with their clinical practice is a normalised part of their career pathway from the beginning of their training. In contrast, gaining a PhD is not a usual part of NMAHPs' career ladder and, unlike the IAT pathway, NMAHPs embark on PhDs only after completing clinical training. This suggests not only that their potential is stymied,

with a resultant delay to benefits emanating from their research,[12] but with fewer years left before retirement, the mantra of 'investment in research leaders of the future'[6] is questionable.

Rather than just focusing on barriers to clinical academic careers, this study further noted the importance of motivation,[5 7] particularly in terms of improving patient care. Findings revealed that early experiences of research were also influential, confirming previous studies that medical trainees who enjoyed research were likely to continue to be motivated,[4 6 8 10] whereas negative experiences could result in people abandoning the clinical academic pathway. However, differences in provision between the two research populations reinforced recommendations that training in research design and evidence evaluation be made available to all healthcare professionals, starting at undergraduate level.[6 25] If implemented, research could become normalised as part of NMAHPs' career pathways, in line with that of MCAs.

MCAs benefited from dedicated academic time and financial support for career development through bursaries and fees. In contrast, NMAHPs often had to leave their post or reduce their hours to pursue clinical academic training. This confirms the findings of the Academy of Medical Sciences who highlight a 'critical need' to develop 'a sustainable infrastructure for research' as well as clearer academic career pathways for NMAHPs, 'supported by increased funding at both predoctoral and postdoctoral levels'.[6, P19]

While MCAs' early experiences of disseminating research and grant applications potentially improved their career prospects, it resulted in delays to their clinical training with subsequent financial implications as well as presenting challenges in managing two sets of training simultaneously. This was a common theme across both data sets, with participants experiencing difficulties negotiating time for mandatory academic work with clinical managers. This echoes previous research suggesting a lack of understanding regarding the purpose and value of clinical academic careers and indicating differences in clinical and academic priorities of HEIs and NHS.[10] This study supports Westwood *et al*'s call for better communication to enable HEIs and NHS partners to work together to smooth the clinical academic pathway.[26]

Moreover, while MCAs had more job security and seemingly better career prospects, their progress, like NMAHPs', was dependent on securing grants and the availability of suitable positions. Participants from both groups likened the clinical academic pathway to a 'pyramid', with positions becoming scarcer and more competitive as they progressed which was considered particularly problematic for people who were limited geographically such as women with caring responsibilities. This study's findings therefore support recent recommendations to create more clinical academic posts, particularly at senior levels[6 13] in order to promote equality and inclusivity and build capacity in clinical research leadership. Furthermore, incorporating research time into clinical academic

job descriptions could attract staff, increase job satisfaction and contribute to improved patient outcomes.[6]

## Strengths and limitations

Combining survey and interview data enhanced the research, with similar themes identified across both data sets. Furthermore, individual interviews enabled a deeper understanding of clinical academic trainees' experiences, allowing exploration of nuanced meanings and apparent inconsistencies. These data can be difficult to capture in surveys or focus groups which are the usual methods for researching clinical academic careers.[4 5 8 10]

Although limited geographically, the study nevertheless encompassed two different HEIs and four large Trusts, revealing important differences and similarities between the clinical academic experiences of MCAs and NMAHPs as well as between two training sites. Suggestions for future research include comparing experiences of clinical academics from different geographical areas and exploring the perspectives of clinical managers.

## CONCLUSION

Although both MCAs and NMAHPs are encouraged to pursue a clinical academic career, this research reveals inconsistencies in their respective career pathways. However, despite perceptions that MCAs have an easier route, this research has revealed obstacles in terms of financial penalisation and difficulties in climbing the clinical academic career ladder for both sets of participants in this study.

Our research suggests that the NIHR could fulfil their aim to 'attract, develop and retain the best research professionals'[1] by providing more well-funded posts at senior level, research skills development in nursing and midwifery undergraduate courses and clearer run-through pathways in nursing/midwifery. Furthermore, academic institutions need to work more closely with clinical employers to ensure academic commitments are properly ring-fenced, protected, and valued. These steps would enable clinical academics from all backgrounds to fully use their dual skills and talents.

**Acknowledgements** The authors wish to thank all of the survey respondents and interview participants in this study. Also, thanks go to Dr Louise Bramley for her contributions, and to the reviewers' for their valuable feedback on a previous version of the manuscript.

**Contributors** All authors (DT, ER and JB) contributed to the study design and data analysis. DT collected the data and drafted the manuscript as lead author. ER and JB contributed comments and edits to the manuscript. Final approval was given by all authors.

**Funding** This work was supported by the National Institute for Health Research (NIHR) Applied Research Collaboration East Midlands (ARC EM) grant number NIHR200171.

**Disclaimer** The views expressed are those of the authors and not necessarily those of the NIHR or the Department of Health and Social Care.

**Competing interests** None declared.

**Patient consent for publication** Not required.

**Ethics approval** Following Health Research Authority guidance, this research was not submitted for Ethics Board approval because participants were recruited

by virtue of their participation in educational programmes, not their NHS status. Nevertheless, the research was conducted in accordance with the Declaration of Helsinki. Any identifying information given in order to enter the prize draw and/or offer to be interviewed, was removed prior to analysis. Interviewees were informed about the purpose of the research, their right to withdraw from the study and how their (anonymised) data would be used. Recorded verbal consent was obtained prior to each interview. When interviewing in a public space, privacy was ensured to maintain confidentiality. In addition, participants were assured that data would be stored confidentially on secure University systems.

**Provenance and peer review** Not commissioned; externally peer reviewed.

**Data availability statement** Data are available on reasonable request. Raw transcripts are held by the authors who will consider requests for further information.

**ORCID iD**
Diane Trusson http://orcid.org/0000-0002-6995-1192

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
