## [Reviewer comments · BMJ Open]

ARTICLE DETAILS

TITLE (PROVISIONAL)	A multi-methods study comparing the experiences of medical clinical academics with nurses, midwives and allied health professionals pursuing a clinical academic career.
AUTHORS	Trusson, Diane; Rowley, Emma; Barratt, Jonathan

VERSION 1 – REVIEW

REVIEWER	Connie Berthelsen University of Southern Denmark Denmark
REVIEW RETURNED	01-Oct-2020

GENERAL COMMENTS	Thank you for the opportunity to read your paper, which is a highly relevant topic. I do however, have some comments and questions: ABSTRACT Please mention mixed methods in the design paragraph of the abstract INTRODUCTION A more thorough presentation of the NMAHP's career pathways and academic work in clinical practice is needed in the introduction, as a contrast to the thick descriptions of the MCA's pathway. We need to be able to see the differences between the clinical academic pathways of the two groups. The aim of the study must be stated in the end of the introduction There are too many abbreviations in the introduction and AHP is not described METHODS Overall the method section is unstructured and leaves many questions: • In the title you claim that this is a mixed methods study. If it is indeed the design and/or methodological approach in your research study – this needs to be explained thoroughly in the study design section of your paper: Which mixed methods design you used, why you used is, how it was used, and the benefits the study had by using mixed methods. Mixed methods as study design also needs to be mentioned in the design paragraph of the abstract and anywhere relevant.• In “Study design” you explain the how the survey was developed and how the data collection differed, in both survey and interviews, for the two groups. Firstly, this needs to be in the data collection
---

	section. "Study design" needs to explain the study design. Survey and interviews are data collection methods – not design.  o What validated questionnaire(s) did you use/ or how did you develop the survey questions? Where they based on literature? o How was the interview guide developed? Based on what? o Why were the participants not given the same survey questions? And how can you then compare and integrate the results?  • In the first section of "Recruitment" you describe how the survey was sent out – this is not recruitment but data collection, where it needs to be moved to. • How can participant 1-5 and 8-9 be both a nurse and a midwife? Do these participants have both educations? • In "Ethical considerations" there needs to be a paragraph about your considerations about interviewing in a public café. • In your analysis section, you write: "For this article, survey and interview data from both studies were combined". o My question is how? Please describe and elaborate how the combination of data was performed according to mixed methods methodology. Was it a triangulating convergence model or?? "Themes emerged from the data through an inductive process involving multiple readings of the data. This reflexive process is used as an iterative way of developing insight" o Please name the qualitative analysis you used, describe the qualitative analysis you used and elaborate the analysis process. • Furthermore, you need to describe how you handled and analysed the quantitative data from the survey. Which statistical analysis did you perform, and how, and which program did you use.
--	---

REVIEWER	Kathleen Leedham-Green Imperial College London I am author of this paper: https://bmjopen.bmj.com/content/10/3/e033480.info
REVIEW RETURNED	02-Oct-2020

GENERAL COMMENTS	Overall, I thought this was a well written and conducted piece of research. It would be greatly enhanced with a figure or table summarising your findings. The process of condensing so many survey responses and interviews could be clarified - were they just read and discussed, or did you use a systematic process? You need an ethics statement re Declaration of Helsinki or some other ethics guidelines. Abstract: well written in plain English. Good number of participants, appropriate study design. Well described context, strengths and limitations. Well defined territory, niche and justification for how your research intends to fill that niche. Good attention to quality criteria.
---

	Note: Bristol Online Survey have changed their name. Needs explanation for international audience. Perhaps call it an academic online survey platform (JISC Online Survey). I wonder whether you need to make more explicit how the gender disparity between nurses/midwives and MCAs makes the out-of-hours academic culture particularly damaging for this group. You make a good case for the pay disparities making extended training and deferral of higher grade salary more difficult for NMAPHs. Research team: Good mix. Is the clinical academic a MCA or NMAPH? Ethics: I'm surprised that this study didn't have ethical review by the institution hosting the research given that participants are being interviewed in their academic rather than clinical capacity. Your ethical statement is however comprehensive and appears to follow the COPE/BERA guidelines. Even if the HRA is not interested, this is still research involving human participants. A statement relating to declaration of Helsinki is needed. I find this resource helpful https://www.mededpublish.org/Policies/Protection-of-Research-Participants#No%20approval As an experienced ethics reviewer, I think a declaration should permit publication. There are ethical implications to not publishing research of this quality. I found your system for numbering respondents confusing and difficult to follow and I was curious about gender. I would prefer something more descriptive (SR1, nurse, female) or (CS1, clinical lecturer, male). I realise you are struggling at the edge of your wordcount, but perhaps the editor could accommodate this in the interests of clarity? Analysis Did you use qualitative software? If not, what was your coding process - did you do this collaboratively, individually, or all by one person? How did you treat minor themes. You have 'parked' themes relating to managing family life and clinical academia. Is this wise as more nurses and midwives are more likely to be primary carers. Perhaps just mention your main themes within this category and say that the richness of the data was such that it will be reported in full separately. This paper would be greatly enhanced by a table or figure summarising your coding tree / or representing your findings in a concept map. I wanted to see an overarching framework to hang the results on as I read. Loved the open start to the interviews. Really important to pick up unknown unknowns. You implications & applications section was well grounded in your data: more well funded posts at senior level, research skills development in nursing & midwifery undergraduate courses, clearer run-through pathways in nursing/midwifery, protected time for research within clinical contracts negotiated with clinical managers. Perhaps move these to your conclusions section? e.g. Our research suggest that NIHR could fulfill their aim to... by...
--	--

	Implications for future research: 'should' is a bit strong. Perhaps 'Suggestions for future research include ...'. References: what is reference 11? 13,14 Your website references need to be appropriately formatted. 22, 23, 24, 25 you need to reference these as reports rather than just website URLs
--	--

VERSION 1 – AUTHOR RESPONSE

Reviewer: 1
Reviewer Name

Connie Berthelsen

Institution and Country

University of Southern Denmark
Denmark

Please state any competing interests or state 'None declared':
None

Comments to the Author

Thank you for the opportunity to read your paper, which is a highly relevant topic. I do however, have some comments and questions:

Response: Thank you for this encouragement. Our responses to each point are shown below.

ABSTRACT

Please mention mixed methods in the design paragraph of the abstract

Response: On reflection, the authors have decided that the term 'mixed methods' is not an accurate description of the methods used in the study (see below for more detail). The design paragraph of the abstract has been amended to read, 'A multi-methods approach was used to elicit qualitative data.'

INTRODUCTION

A more thorough presentation of the NMAHP's career pathways and academic work in clinical practice is needed in the introduction, as a contrast to the thick descriptions of the MCA's pathway. We need to be able to see the differences between the clinical academic pathways of the two groups.

Response: An illustrative chart (figure 2) has been added which shows the NMAHP clinical academic pathway more clearly.

The aim of the study must be stated in the end of the introduction

Response: The final sentence of the introduction now reads, 'The aim is to compare their experiences of pursuing a clinical academic career between the two study populations.'

There are too many abbreviations in the introduction and AHP is not described

Response: Although the authors agree that there are a lot of abbreviations in the introduction, we feel that most are necessary, apart from MB and IATP, which have been removed. AHP has been described in full.

METHODS

Overall the method section is unstructured and leaves many questions:

- In the title you claim that this is a mixed methods study. If it is indeed the design and/or methodological approach in your research study – this needs to be explained thoroughly in the study design section of your paper: Which mixed methods design you used, why you used it, how it was used, and the benefits the study had by using mixed methods. Mixed methods as study design also needs to be mentioned in the design paragraph of the abstract and anywhere relevant.

Response: On reflection, the authors have realised that describing the study as mixed methods is misleading, since quantitative data analysis was not applied (i.e. no statistical analyses were performed). Rather, the survey data that the study is based on is from free text responses. We have decided that multi-methods would be a more suitable description. The study design has been altered to discuss why a qualitative approach was used and how. Also, all references to mixed methods have been removed.

- In “Study design” you explain the how the survey was developed and how the data collection differed, in both survey and interviews, for the two groups. Firstly, this needs to be in the data collection section. “Study design” needs to explain the study design. Survey and interviews are data collection methods – not design.

Response: The description of the survey development and data collection have been moved to the data collection section. The study design section now focusses on the qualitative methods that were used in the study.

o What validated questionnaire(s) did you use/ or how did you develop the survey questions? Were they based on literature?

Response: We have added a description of how the survey questions were developed, based on the literature.

o How was the interview guide developed? Based on what?

Response: We have added a description of how the interview guides were developed, based on the literature and responses to the surveys.

o Why were the participants not given the same survey questions? And how can you then compare and integrate the results?

Response: We have clarified that the survey questions were the same for all participants, apart from some questions specific to their profession. We have also explained that an additional question in the MCA survey was in response to a theme which had arisen in the NMAHP study.

- In the first section of “Recruitment” you describe how the survey was sent out – this is not recruitment but data collection, where it needs to be moved to.

Response: The description of how the survey was sent out has been moved to the ‘data collection’ section.

- How can participant 1-5 and 8-9 be both a nurse and a midwife? Do these participants have both educations?

Response: We apologise for this misleading information. The correct job titles are now shown in table 1

- In “Ethical considerations” there needs to be a paragraph about your considerations about interviewing in a public café.

Response: The following statement has been added: ‘When interviewing in a public space, privacy was ensured to maintain confidentiality.’

- In your analysis section, you write:

“For this article, survey and interview data from both studies were combined”.

o My question is how? Please describe and elaborate how the combination of data was performed according to mixed methods methodology. Was it a triangulating convergence model or??

“Themes emerged from the data through an inductive process involving multiple readings of the data. This reflexive process is used as an iterative way of developing insight”

Response: This section has been reworked to explain in more detail how the data were combined and analysed using a framework analysis, and how the themes emerged. A coding tree (figure 5) has been added in response to reviewer 2’s suggestion.

o Please name the qualitative analysis you used, describe the qualitative analysis you used and elaborate the analysis process.

Please see our response to the previous point.

- Furthermore, you need to describe how you handled and analysed the quantitative data from the survey. Which statistical analysis did you perform, and how, and which program did you use.

Response: No statistical analysis was performed on the survey data, hence the removal of the term mixed methods in the revised document.

Reviewer: 2

Reviewer Name

Kathleen Leedham-Green

Institution and Country

Imperial College London

Please state any competing interests or state ‘None declared’:

I am author of this paper: <https://bmjopen.bmj.com/content/10/3/e033480.info>

Comments to the Author

Overall, I thought this was a well written and conducted piece of research. It would be greatly enhanced with a figure or table summarising your findings. The process of condensing so many survey responses and interviews could be clarified - were they just read and discussed, or did you use a systematic process? You need an ethics statement re Declaration of Helsinki or some other ethics guidelines.

Response: The authors are grateful for this encouragement. Please see our responses below to each of the points raised.

Abstract: well written in plain English.

Good number of participants, appropriate study design.

Well described context, strengths and limitations.

Well defined territory, niche and justification for how your research intends to fill that niche.

Good attention to quality criteria.

Note: Bristol Online Survey have changed their name. Needs explanation for international audience. Perhaps call it an academic online survey platform (JISC Online Survey).

Response: Thank you for bringing this to our attention. The name of the survey has been amended, along with the corresponding references.

I wonder whether you need to make more explicit how the gender disparity between nurses/midwives and MCAs makes the out-of-hours academic culture particularly damaging for this group. You make a good case for the pay disparities making extended training and deferral of higher grade salary more difficult for NMAPHs.

Response: The gender disparity in relation to caring responsibilities will be discussed in a forthcoming paper on the topic of combining clinical academic careers with family life. This has been indicated in the current manuscript.

Research team: Good mix. Is the clinical academic a MCA or NMAPH?

Response: Thank you. A sentence has been added to explain that a clinical academic nurse was part of the team for the NMAHP study, and a clinical academic surgeon was part of the team for the MCA study.

Ethics: I'm surprised that this study didn't have ethical review by the institution hosting the research given that participants are being interviewed in their academic rather than clinical capacity. Your ethical statement is however comprehensive and appears to follow the COPE/BERA guidelines. Even if the HRA is not interested, this is still research involving human participants. A statement relating to declaration of Helsinki is needed. I find this resource helpful
<https://www.mededpublish.org/Policies/Protection-of-Research-Participants#No%20approval>
As an experienced ethics reviewer, I think a declaration should permit publication. There are ethical implications to not publishing research of this quality.

Response: Thank you for the suggested resource. The following sentence has been added:
'Nevertheless, the research was conducted in accordance with the Declaration of Helsinki.'

I found your system for numbering respondents confusing and difficult to follow and I was curious about gender. I would prefer something more descriptive (SR1, nurse, female) or (CS1, clinical lecturer, male). I realise you are struggling at the edge of your word count, but perhaps the editor could accommodate this in the interests of clarity?

Response: The system for numbering respondents has been altered, and the gender of each participant quoted has been entered after their citation.

Analysis

Did you use qualitative software? If not, what was your coding process - did you do this collaboratively, individually, or all by one person? How did you treat minor themes. You have 'parked' themes relating to managing family life and clinical academia. Is this wise as more nurses and midwives are more likely to be primary carers. Perhaps just mention your main themes within this category and say that the richness of the data was such that it will be reported in full separately.

Response: The coding process has been clarified. Also, the plan to discuss the rich data in relation to managing family life and clinical academia in a separate paper (which is currently being prepared), has been stated more explicitly.

This paper would be greatly enhanced by a table or figure summarising your coding tree / or representing your findings in a concept map. I wanted to see an overarching framework to hang the results on as I read.

Response: A coding tree has been created to clarify the major themes (figure 6).

Loved the open start to the interviews. Really important to pick up unknown unknowns.

Response: Thank you for this encouraging comment!

Your implications & applications section was well grounded in your data: more well-funded posts at senior level, research skills development in nursing & midwifery undergraduate courses, clearer run-through pathways in nursing/midwifery, protected time for research within clinical contracts negotiated with clinical managers. Perhaps move these to your conclusions section? e.g. Our research suggest that NIHR could fulfill their aim to... by...

Response: Thank you for this suggestion. The additional wording has made the conclusion much stronger.

Implications for future research: 'should' is a bit strong. Perhaps 'Suggestions for future research include ...'.

Response: The section has been reworded as suggested.

References: what is reference 11?

Response: Reference 11 was anonymised in error but now shows the correct reference.

13,14 Your website references need to be appropriately formatted.

22, 23, 24, 25 you need to reference these as reports rather than just website URLs

Response: All of the references have been amended (although the numbers are different after the addition of new references)

VERSION 2 – REVIEW

REVIEWER	Kathleen Leedham-Green
----------	------------------------

	Medical Education Research Unit, Imperial College London
REVIEW RETURNED	25-Jan-2021

GENERAL COMMENTS	Thank you for submitting this revised version. The content is important and I look forward to reading the published version. I have a very few tiny niggles, and I am happy to recommend this paper once these are addressed without needing to see the revised version (line numbers refer to the margin numbers, not the actual line): Once someone is post-doctoral is their academic work still considered 'training'? If not, page 3 line 60 should read 'enabling postdoctoral clinicians to split their time equally between clinical and academic work'. Full stop p5 L60. P6 L3 change to "indicated that they were willing..." or "indicated their willingness..." P10 l3, use colons in headings consistently or not at all Rigour Rather than saying this increased the credibility, or reliability, I would just say which steps you have taken and allow the reader to judge. This paragraph should be part of your methods and probably doesn't need a separate title - all methods should be rigorous. Patient and public involvement Not sure how well this paragraph works. 'There was no patient or public involvement in this study' should suffice. You have already said how the research team have insights into the subject of interest in your paragraph on the Research Team (NB check your heading capitalization is consistent throughout). P24 L18 I'd caveat any assertion not directly supported by your data. So rather than saying 'incorporating research time into clinical academic job descriptions would attract staff...' perhaps use 'could attract staff' or 'has the potential to attract staff' You have two sections on strengths and limitations. These need to be merged. I would add a sentence to your conclusions on the need for academic institutions to work more closely with clinical employers to ensure academic commitments are properly ring-fenced/protected/valued. I'm not sure what table 1 adds to figure 1 - can they be combined? For the international audience, I think you need to reduce the number of abbreviations and acronyms. Even ones like AFP, AHP, ACF, HEE, CL and HEI might not be familiar outside the UK. It interrupts the flow for the reader, having to keep going back to find these definitions. If you want to include them, then perhaps a list of abbreviations at the end is needed. I'd personally advocate dropping them and using the full term, but I appreciate that impacts on your word count.
---